# MC-LR Aggravates Liver Lipid Metabolism Disorders in Obese Mice Fed a High-Fat Diet via PI3K/AKT/mTOR/SREBP1 Signaling Pathway

**DOI:** 10.3390/toxins14120833

**Published:** 2022-11-30

**Authors:** Hanyu Chu, Can Du, Yue Yang, Xiangling Feng, Lemei Zhu, Jihua Chen, Fei Yang

**Affiliations:** 1Hunan Province Key Laboratory of Typical Environmental Pollution and Health Hazards, School of Public Health, University of South China, Hengyang 421001, China; 2Xiangya School of Public Health, Central South University, Changsha 410078, China; 3School of Public Health, Changsha Medical University, Changsha 410219, China; 4The Key Laboratory of Ecological Environment and Critical Human Diseases Prevention of Hunan Province, Department of Education, School of Basic Medical Sciences, Hengyang Medical School, University of South China, Hengyang 421001, China

**Keywords:** MC-LR, aggravates, liver, lipid metabolism, obese mice

## Abstract

Obesity, a metabolic disease caused by excessive fat accumulation in the body, has attracted worldwide attention. Microcystin-LR (MC-LR) is a hepatotoxic cyanotoxin which has been reportedly to cause lipid metabolism disorder. In this study, C57BL/6J mice were fed a high-fat diet (HFD) for eight weeks to build obese an animal model, and subsequently, the obese mice were fed MC-LR for another eight weeks, and we aimed to determine how MC-LR exposure affects the liver lipid metabolism in high-fat-diet-induced obese mice. The results show that MC-LR increased the obese mice serum aspartate aminotransferase (AST) and alanine aminotransferase (ALT), indicating damaged liver function. The lipid parameters include serum triglyceride (TG), total cholesterol (TC), low-density lipoprotein cholesterol (LDL-c), and liver TG, which were all increased, whilst the high-density lipoprotein cholesterol (HDL-c) was decreased. Furthermore, after MC-LR treatment, histopathological observation revealed that the number of red lipid droplets increased, and that steatosis was more severe in the obese mice. In addition, the lipid synthesis-related genes were increased and the fatty acid β-oxidation-related genes were decreased in the obese mice after MC-LR exposure. Meanwhile, the protein expression levels of phosphorylation phosphatidylinositol 3-kinase (*p*-PI3K), phosphorylation protein kinase B (*p*-AKT), phosphorylation mammalian target of rapamycin (*p*-mTOR), and sterol regulatory element binding protein 1c (SREBP1-c) were increased; similarly, the *p*-PI3K/PI3K, *p*-AKT/AKT, *p*-mTOR/mTOR, and SREBP1/β-actin were significantly up-regulated in obese mice after being exposed to MC-LR, and the activated PI3K/AKT/mTOR/SREBP1 signaling pathway. In addition, MC-LR exposure reduced the activity of superoxide dismutase (SOD) and increased the level of malondialdehyde (MDA) in the obese mice’s serum. In summary, the MC-LR could aggravate the HFD-induced obese mice liver lipid metabolism disorder by activating the PI3K/AKT/mTOR/SREBP1 signaling pathway to hepatocytes, increasing the SREBP1-c-regulated key enzymes for lipid synthesis, and blocking fatty acid β-oxidation.

## 1. Introduction

With the improvement in people’s living standards and economic situations, obesity has emerged as one of the world’s major health problems. In the past 30 years, its prevalence has more than doubled globally. According to the World Health Organization (WHO) survey data in 2016, there are more than 1.9 billion adults worldwide who are overweight or obese, with a 38.9% incidence rate [1]; as the fifth leading cause of death worldwide, obesity and overweight are responsible for about 1.3 million fatalities annually [2]. Obesity is characterized as an abnormal or excessive buildup of lipids in the body that can be attributed to various causes, including a sedentary lifestyle, overeating, and genetics [3]. Several studies have shown strong correlation between obesity and a higher risk of type 2 diabetes, hyperglycemia, coronary heart disease, renal disease, and non-alcoholic fatty liver (NAFLD) [4,5,6]. The long-term intake of a high-fat diet (HFD) will increase the risk of becoming obese, cause an imbalance between energy intake and consumption, increase and release free fatty acids into the blood, and finally result in a disturbance of lipid metabolism [7,8,9].

The frequency of cyanobacteria blooms increased in recent years as a result of global warming, increased nitrogen, phosphorus emissions, and the deepening of water eutrophication [10,11,12]. The cyanobacterial toxin produced by cyanobacteria blooms causes considerable harm to the aquatic environment [13,14]. Microcystins (MCs) are a class of cyclic heptapeptide toxins [15], among which approximately 250 different structural analogs were found. One of the most common, strongest toxins, and frequently studied is the Microcystin-LR (MC-LR) [16]; it can enter the human body by drinking water, skin contact, or aquatic products, and then accumulate in the liver, kidney, intestine, brain, lung, heart, immune, and reproductive system, so as to produce the corresponding toxic effects which are hazardous to wildlife and human health [17,18,19,20,21,22,23,24,25]. MC-LR is difficult to decompose and cannot be eliminated by a common water treatment process [12,26,27,28,29]. The WHO established the guideline values for MC-LR in drinking water not exceeding 1 μg/L in 2020 [30]. According to the International Agency for Research on Cancer (IARC), MC-LR was defined as a group 2B carcinogen [31]. The liver is the target organ of MC-LR, and it has been well-known that the hepatotoxicity of MC-LR is primarily caused by its specific and intense inhibition of intracellular serine/threonine phosphatases 1 and 2A (PP1 and PP2A), production of reactive oxygen species, disruption of cytoskeleton, and inflammation [32,33,34].

The liver is the main organ whereby the human body performs lipid metabolism. Numerous studies have shown that MC-LR exposure can cause liver lipid metabolism disorder and many liver diseases [17,35]. Arman et al. demonstrated that MC-LR (10 g/kg, 30 g/kg) exposure for 4 weeks in Sprague Dawley rats drives the high-fat/high-cholesterol (HFHC) diet-induced NAFLD in the liver towards a more severe phenotype [36]. In fishers who have lived near Lake Taihu for many years, the indices of aspartate aminotransferase (AST)/alanine aminotransferase (ALT), triglyceride (TG), and total cholesterol (TC) were elevated after chronic MC exposure, leading to liver damage and lipid metabolism dysfunction [37]. Subsequently, mice were long-term exposed to environment-related concentrations of MC-LR, which resulted in liver lipid metabolism disorder, including increased lipid production and decreased lipid β-oxidation [17,37]. Fatty acid β-oxidation and the output of hepatic lipoprotein were significantly inhibited in mice following short-term exposure to low-dose MC-LR, resulting in non-alcoholic steatohepatitis (NASH) [38]. Steatosis was also found in the livers of mice treated with 50 and 100 μg/kg MC-LR for one month [39]. After male zebrafish underwent MC-LR (0, 1, 10 μg/L) treatment for 90 days, which caused endoplasmic reticulum stress and mitochondrial malfunction, this resulted in a disturbance of hepatic lipid metabolism [40]. After 28 days of intragastric injection, cross-omics technologies (metabonomics, transcriptomics, and metagenomics) revealed that MC-LR might activate peroxisome proliferator-activated receptors (PPARs) and caused problems in the liver lipid metabolism of mice livers [41].

However, little research has reported on the effects of MC-LR on HFD-induced obese mice liver and the preliminary mechanism. This study aimed to investigate the effects of MC-LR on hepatic lipid metabolism in obese mice fed a high-fat diet.

## 2. Results

### 2.1. Effects of MC-LR on Body Weight, Food Intake, and Liver Index in HFD-Induced Obese Mice

In the last weeks of the experiment, visual observation showed that the body weights of the HFD- and MC-LR+HFD-treated mice had significantly increased compared to the control mice (Figure 1A), and during the experiment, HFD- and MC-LR+HFD-treated mice had gained significantly more weight than the control mice, while there were no differences in the overall trend between the HFD- and MC-LR+HFD-treated mice (Figure 1B). The food intake of the HFD and MC-LR+HFD groups was reduced in 2, 10, 13, and 14 weeks compared to the control mice, respectively. Furthermore, after MC-LR treatment, the food intake was reduced in 9 and 10 weeks (Figure 1C). Mice treated with HFD and MC-LR+HFD had significantly increased liver tissue weights compared with the control mice (Figure 2A–C).

### 2.2. Biochemical Indexes

As shown in Figure 3, the serum ALT and AST were all increased compared with the control group, and the levels of serum ALT and AST were remarkably increased in the HFD+MC-LR group compared with the HFD-treated mice. In order to evaluate the effect of MC-LR on lipid metabolism in high-fat-diet-induced obese mice, the serum and hepatic lipid levels of each group were measured, and the results are shown in Figure 4. The results show that the serum TG, TC, and low-density lipoprotein cholesterol (LDL-c) levels in the HFD and HFD+MC-LR groups were significantly increased compared to those in the control group. Furthermore, the level of high-density lipoprotein cholesterol (HDL-c) in serum showed a significant reduction trend in the HFD+MC-LR group in comparison with that of the control group. Compared with the HFD-treated mice, the serum TG, TC, and LDL-C levels were significantly increased in the HFD+MC-LR group. However, there was no significant difference in serum HDL-c levels between the HFD and HFD+MC-LR groups (Figure 4A–D). The liver TG and TC contents in the HFD and HFD+MC-LR groups were significantly increased compared to the control group. Similarly, the hepatic TG and TC levels were increased in HFD+MC-LR mice compared with the HFD mice (Figure 4E,F).

### 2.3. Pathological Observation of Liver

Using H&E and oil red O to stain the liver histological slices, we found no alterations in the control group whilst the hepatocytes were uniform in size and closely arranged. Compared with the normal control group, the lipid droplets were increased in HFD and HFD+MC-LR-treated mice. In addition, it can be observed that the number of fat vacuoles and red lipid droplets in the HFD+MC-LR mice appears to be greater than in HFD mice, and the degree of steatosis of the liver tissue in the HFD+MC-LR group was more serious. MC-LR could significantly aggravate the lipid accumulation in the liver of obese mice (Figure 5).

### 2.4. Effects of MC-LR on Lipid Synthesis and Fatty Acid β-Oxidation in Liver 

To determine whether MC-LR aggravates the liver lipid metabolism disorder in obese mice induced by a high-fat diet, the relative mRNA expression levels of lipid synthesis-related genes and fatty acid β-oxidation-related genes were detected by qPCR. Compared with the control group, the expression level of lipid synthesis-related genes’ sterol regulatory element-binding transcription factor-1c (SREBP-1c), acetyl coenzyme a carboxylase 1 (ACC1), fatty acid synthase (FASN), cluster of differentiation 36 (CD36), peroxisome proliferator-activated receptor gamma (PPAR-γ), stearoyl-CoA desaturase-1 (SCD1), diacylglycerol O-acyltransferase 1 (DGAT1), and diacylglycerol O-acyltransferase 2 (DGAT2) were all up-regulated in HFD and HFD+MC-LR-treated groups. In addition, compared with the HFD mice group, these lipid synthesis-related genes were all significantly increased in HFD+MC-LR group (Figure 6A–H).

The mRNA levels of the fatty acid β-oxidation-related genes carnitine palmitoyltransferase I (CPT-1α) and peroxisome proliferator-activated receptor alpha (PPAR-α) were reduced in the HFD mice group compared with the control. Importantly, the levels of CPT-1α and PPAR-α were evidently decreased (Figure 6I,J).

### 2.5. Effect of MC-LR on PI3K/AKT/mTOR/SREBP1 Pathway in Liver

In order to further explore the molecular mechanism whereby MC-LR aggravates liver lipid metabolism disorders in obese mice fed a high-fat diet, the WB method was used to detect the protein expression levels of phosphoinositide 3-kinase (PI3K), phosphorylation phosphatidylinositol 3-kinase (*p*-PI3K), protein kinase B (AKT), phosphorylation protein kinase B (*p*-AKT), mammalian target of rapamycin (mTOR), phosphorylation mammalian target of rapamycin (*p*-mTOR), and sterol regulatory element binding protein 1c (SREBP1-c). As shown in Figure 7A, the nuclear SREBP1-c significantly increased in the HFD and HFD+MC-LR groups compared with the control mice; similarly, compared with the HFD mice, the expression level of nuclear SREBP1-c was significantly increased. The expression of SREBP1-c, *p*-PI3K/PI3K, *p*-AKT/AKT, and *p*-mTOR/mTOR proteins were all increased in the HFD and HFD+MC-LR treatment groups (Figure 7B). These results indicate that MC-LR aggravates liver lipid metabolism disorders in obese mice fed a high-fat diet by activating the PI3K/AKT/mTOR/SREBP1 pathway.

### 2.6. Effects of MC-LR on Serum Oxidative Stress Status in HFD-Induced Obese Mice

As shown in Figure 8A, the level of serum malondialdehyde (MDA) was increased in the HFD and HFD+MC-LR groups in comparison with control mice, and the MDA concentration in the HFD+MC-LR group was significantly higher than in HFD mice. In addition, the activity of serum superoxide dismutase (SOD) was reduced in the HFD+MC-LR group compared with the control and HFD groups (Figure 8B).

## 3. Discussion

### 3.1. Effects of MC-LR on General Conditions in Obese Mice

In this study, HFD-fed mice had a higher body weight. After MC-LR treatment, we found no significant changes in body weight and no significant changes in overall trends in food consumption. Similarly, the mice treated with HFD and MC-LR+HFD had increased absolute and relative liver tissue weights. These results show that exposure to MC-LR had little effect on the obese mice’s body and liver weights. 

### 3.2. MC-LR Aggravates Liver Damage in Obese Mice

As AST and ALT are essential indicators reflecting liver function, the above indexes increase when the hepatocytes are damaged, and the degree of increase is positively correlated with the degree of hepatocyte damage [42]. Compared with the control group, the activities of serum AST and ALT were increased in mice, which indicates that the liver has been damaged. MC-LR might cause histopathological damage in mice liver tissue, such as the disordered arrangement of hepatic lobules with obvious inflammatory cell infiltration and liver steatosis [17,43]. After exposure to MC-LR, the serum AST and ALT in obese mice were still increased. In the NAFLD animal model, MC-LR exposure significantly increased the ALT level [35]. In addition, following exposure to MC-LR, the numbers of fat vacuoles and red lipid droplets increased in obese mice, and the degree of steatosis of liver tissue was more serious. These results indicate that MC-LR aggravates the damage to the obese mice livers. 

### 3.3. MC-LR Aggravates Liver Lipid Metabolism in Obese Mice

In this study, the decreased serum HDL-c and the increased serum TG, TC, LDL-c, and liver TG and TC were all induced by MC-LR, which was consistent with our previous results [17]. In addition, histopathologically, here, the hepatocytes of HFD-fed mice presented increased red lipid droplets, leading to excessive lipid accumulation. These results suggest that MC-LR notably enhances obese mice lipid deposits in hepatocytes.

The lipid metabolism mainly includes lipid synthesis and fatty acid β-oxidation. The de novo synthesis of fatty acids (DNL) is a fundamental biosynthetic pathway within the liver, contributing to the lipids that are stored and secreted by hepatocytes [44].The process in hepatocytes involves a variety of regulatory factors. SREBP1-c is a steroid regulatory element binding protein and the key regulator of endogenous fatty acid synthesis in hepatocytes [45,46,47], which matured in the nucleus and activates the expression of the downstream genes ACC1, FASN, and SCD1, as well as promotes endogenous free fatty acid synthesis. ACC1 is an integral enzyme in de novo lipogenesis [48]. FASN is a key enzyme regulating the de novo fatty acid synthesis pathway. SCD1 catalyzes the rate-limiting step in the conversion of saturated fatty acids to mono-unsaturated fatty acids [49]. The activation of SREBP1-c occurs via the PI3K/AKT pathway, resulting in the phosphorylation of the nascent SREBP1-c itself [50]. Further to this, the mammalian target of rapamycin complex 1 (mTORC1) is an integral part of this pathway, which is activated when constituently active AKT is expressed in hepatocytes, leading to the increased accumulation of the mature form of SREBP1-c and promotes liver lipid synthesis via DNL [51]. The PI3K/AKT/mTOR signaling pathway, including the PI3K/AKT pathway and its primary downstream target mTOR, plays a critical role in regulating lipid metabolism [52,53]. Furthermore, the PI3K/AKT/mTOR pathway could activate the SREBP1-c and regulate lipogenesis [54,55]. Numerous studies have suggested that the deregulation of the PI3K/AKT signaling pathway was responsible for increased de novo lipogenesis and then exacerbated nonesterified fatty acid (FFA) flux to the liver by increasing lipolysis, which finally led to NASH [56,57]. Similarly, a previous study about hepatocellular carcinoma suggested that the activation of the AKT/mTOR pathway could elevate the expression of SREBP1-c and then reprogram hepatic lipid metabolism [58]. It has been shown that MC-LR induces hepatotoxicity by activating the PI3K/AKT signaling pathway. However, there is no study of the role of the PI3K/AKT/mTOR/SREBP1 signaling pathway in obese mice induced by MC-LR. In this study, the protein expression of PI3K, *p*-PI3K, AKT, *p*-AKT, mTOR, and *p*-mTOR were all increased, and the expression of *p*-PI3K/PI3K, *p*-AKT/AKT, and *p*-mTOR/mTOR protein were also increased in the HFD mice after exposure to MC-LR. Our current finding shows that the MC-LR treatment markedly upregulated the PI3K/AKT/mTOR signaling pathway activation in obese mice, promoting lipid synthesis. We suggest that this pathway is involved in the MC-LR aggravation of imbalanced lipid metabolism in obese mice hepatocytes.

In addition, the nuclear SREBP1-c and SREBP1-c were significantly increased in our study, meaning that SREBP1-c entered the nucleus and activated the downstream target genes ACC1, FASN, SCD1 [49]. Tarana et al. found that the MC-LR exposure (30 µg/kg) for six weeks increased the ACC1, FASN, and SCD1 in the HFHC group [36]. The exogenous fatty acids and the endogenous fatty acids from the DNL way can further synthesize triglycerides (TGs). CD36 is an exogenous fatty acid transporter on the cell membrane, which promotes the transport of circulating fatty acids into hepatocytes [59], and the endogenous fatty acids’ synthesis-related enzymes including SCD1, DGAT1, DGAT2, and PPAR-γ [49,60]. DGAT is the only rate limiting enzyme required for the synthesis of TGs and is involved in fat metabolism and the deposition of lipids. DGAT1 is a key enzyme that controls the rate of TG synthesis in adipocytes [61]. DGAT2 is responsible for the balance of basic fat in various tissues, and catalyzes the biosynthesis of triacylglycerol (TAG) by converting diacylglycerol (DAG) and fatty acyl coenzyme A (CoA) into TAG [62]. Our present results confirm that the MC-LR treatment significantly increased the CD36, SCD1, DGAT1, DGAT2, and PPAR-γ levels in obese mice livers, indicating that MC-LR could promote the increased transcription and expression of lipid synthesis in obese mice. Fatty acid β-oxidation in mitochondria is the main means whereby fatty acids consume hepatocytes as CPT-1α is located on the mitochondrial outer membrane, a key rate-limiting enzyme of fatty acid β-oxidation. In addition, PPAR-α also plays an important role in catalyzing the β-oxidation of fatty acids in mitochondria, and enhances the expression level of CPT-1α to promote fatty acid oxidation [63,64]. In this study, after exposure to MC-LR, the expression of fatty acid β-oxidation-related genes PPAR-α and CPT-1α were reduced in obese mice, indicating that the MC-LR exposure might decrease the triglyceride oxidation capacity, and the serum and liver TG significantly increased, leading to excessive lipid accumulation in the liver. These data are congruent with previous results, wherein after sub-chronic exposure to MC-LR in the context of preexisting diet-induced NASH, the level of CPT-1α decreased in mice livers [36]. In our previous investigation, mice were exposed to MC-LR at environmental concentrations for 9 months through drinking water, as a separate MC-LR control group, which found that MC-LR exposure (120 μg/L) enhanced lipid synthesis and decreased fatty acid β-oxidation, leading to lipid accumulation, and induced hepatic lipid metabolism disorder [17]; compared with our previous investigation results, the expression levels of lipid synthesis–related genes were increased in the HFD-induced obese mice in this study. Therefore, this study suggests that MC-LR (120 μg/L) can aggravate imbalanced lipid metabolism in obese mice induced by a high-fat diet promoting lipid synthesis and inhibiting fatty acid β-oxidation.

### 3.4. MC-LR Induced and Aggravated the Serum Oxidative Stress Status

A high-fat diet will increase the free fatty acids in the blood, cause excessive liver fat and a lipotoxic environment, lead to mitochondrial dysfunction and the production of a large number of ROS, and finally cause oxidative stress [7,9]. SOD is a widely existing free radical scavenger in the body which plays a vital role in alleviating oxidative stress in the body [65]. MDA is an aldehyde substance produced by a lipid peroxidation reaction which can reflect the degree of body peroxidation [66]. It was found that MC-LR treatment could reduce the activity of SOD in the serum of obese mice and increase the level of MDA, thus reducing the overall antioxidant level of the body. Mice were observed to have lower SOD activity and higher MDA activity in the liver after prolonged exposure to low oral doses of MC-LR [39]. It has been proven that the suspected activation of liver oxidative stress is closely related to NAFLD [67]. Therefore, the results of this study suggest that MC-LR can cause oxidative stress by reducing the activity of antioxidant enzymes, leading to the production of reactive oxygen species (ROS) and to obese mice developing severe liver disease.

## 4. Conclusions

In summary, this study indicated that MC-LR could aggravate the HFD-induced obese mice liver lipid metabolism disorder and may even cause more severe liver diseases, such as NAFLD or NASH. The mechanism involves MC-LR activating the PI3K/AKT/mTOR/SREBP1 signaling pathway, promoting the up-regulating the lipid synthesis, and inhibiting the fatty acid β-oxidation in obese mice. In addition, MC-LR induced and aggravated the serum oxidative stress status, and may exacerbate the obesity process. Thus, our study may provide novel insights for preventing the effects of the environment risk factor for obesity, namely MC-LR.

## 5. Materials and Methods

### 5.1. Chemicals

MC-LR (purity ≥ 95%) was purchased from Taiwan Algal Science Inc. (Taiwan, China) and stored at −20 °C. Dimethyl sulfoxide (DMSO) (purity of ≥99.7%) was obtained from Sigma-Aldrich (St. Louis, MO, USA). The normal-fat diet and high-fat diet were obtained from Research Diets Inc. (New Brunswick, NJ, USA).

### 5.2. Animals and Diet

Eight-week-old male C57BL/6J mice were obtained from Central South University (Hunan, Changsha, China) and housed under standard laboratory conditions (20–24 °C, 40–60% humidity, and 12 h light–dark cycle). The mice were given one week to acclimate to their new surroundings and unlimited access to water and food. A total of 15 animals were randomly divided into the two following groups: mice fed a normal-fat diet (NFD, 10 kcal% fat, 70 kcal% carbohydrate, 20 kcal% protein) formed the control group (*n* = 5); and all other mice were fed a high-fat diet (HFD, 60 kcal% fat, 20 kcal% carbohydrate, 20 kcal% protein) (*n* = 10) for eight weeks to establish the obesity models. Then, the obese mice in the HFD group were further divided into two groups, namely the HFD group (*n* = 5) and the HFD+MC-LR group (*n* = 5). All three groups (the control group, the HFD group, and the HFD+MC-LR group) of mice were fed a normal-fat diet, a high-fat diet, and the high-fat diet with 120 μg/L MC-LR in drinking water, respectively, for another eight weeks. The mice were individually housed and exposed to MC-LR via drinking water. The water containing MC-LR was replaced once a week. The mice in the NFD and HFD groups were given an equal volume of diet. Their food intake level was recorded every day, and their body weights were measured weekly. All animal experiments were performed in accordance with the protocol (Approval Number: XYGW-2018-41) approved by the Institutional Animal Care and Use Committee of Central South University.

### 5.3. Detection of Biochemical Indexes

After 16 weeks, the blood samples were collected from the femoral artery of the mice and stored overnight at 4 °C. Serum was separated following centrifugation at 4000 g for 30 min. The serum alanine aminotransferase (ALT), aspartate aminotransferase (AST), triacylglycerol (TG), total cholesterol (TC), low-density lipoprotein cholesterol (LDL-c), high-density lipoprotein cholesterol (HDL-c), SOD, and MDA were determined using commercially available assay kits (Jiancheng Bioengineering Institute, Nanjing, China). In addition, liver tissue with normal saline grind (60 HZ, 60 s, 4 °C), homogenates were centrifuged three times to yield supernatants (2500 g, 10 min, 4 °C) for the detection of TG and TC according to the instructions of the kits (Nanjing Jiancheng Bioengineering Institute, Nanjing, China).

### 5.4. Hematoxylin and Eosin and Oil Red O Staining

After the mice were killed by cervical dislocation, the liver tissues were collected, washed with normal saline, and immediately weighed. Then, parts of the livers were fixed with 4% paraformaldehyde and embedded in paraffin, whilst 4 μm-thick sections were cut and stained with hematoxylin and eosin (H&E). Other parts of the liver tissues fixed in 4% paraformaldehyde were embedded at an optimum cutting temperature for the frozen sections, and the sections were stained with Oil Red O. The sections were observed and photographed with the Invert microscope (Motic Group Co., Ltd. Xiamen, China).

### 5.5. Quantitative RT-PCR

Total RNA was isolated from liver tissues using Trizol (Life Technologies, Shanghai, China), and reverse-transcribed into cDNA with HiScript^®^ Ⅱ Q RT SuperMix for qPCR (Vazyme, Nanjing, China). qRT-PCR was performed with a ChamQ Universal SYBR qPCR Master Mix (Vazyme, Nanjing, China) on a qTOWER3 Real-Time PCR System (Analytikjena, Germany) and the conditions were as follows: preincubation at 95 °C for 30 s, 40 cycles of 95 °C for 10 s, 60 °C for 30 s, and 72 °C for 25 s. The primer 5.0 online software was used to design the primers involved in this paper. The PCR primer sequences are shown in Table 1, and the suitability of primers was detected by means of RT-PCR. The results obtained were normalized to the expression level of the housekeeping gene β-actin, and the relative mRNA expression levels were determined by the 2^−△△Ct^ method.

### 5.6. Western Blot 

Total protein and nuclear protein were isolated from the liver tissues by RIPA buffer (Beyotime, Shanghai, China) and Nuclear Extraction Kit (Invent, Erlangen, Germany), respectively. The protein concentration was determined by the BCA method (Beyotime, Shanghai, China). Proteins were separated using 10% sodium dodecyl sulphate–polyacrylamide gel electrophoresis (SDS-PAGE) and transferred to the polyvinylidene fluoride (PVDF) membranes (Merck Millipore Ltd., Darmstadt, Germany). Next, the transferred membranes were blocked using Protein Free Rapid Blocking Buffer (EpiZyme Biotechnology, Shanghai, China). Membranes were incubated with primary specific antibodies at 4 °C overnight. Primary specific antibodies included anti-SREBP-1c, anti-PI3K, anti-*p*-PI3K, anti-AKT, anti-*p*-AKT, anti-mTOR, anti-*p*-mTOR, and anti-β-actin. Then, the PVDF membranes were washed and incubated with goat anti-mouse IgG (H+L) HRP conjugate or goat anti-rabbit IgG (H+L) HRP conjugate at room temperature for 1 h. Finally, Luminata Forte Western HRP substrate (Millipore, Darmstadt, Germany) was added to detect a specific protein expression using the Bio-Rad chemiluminescence imaging system (Bio-Rad, CA, USA). The intensity of the bands was quantitated by ImageJ (Rawak Software Inc., Stuttgart, Germany).

### 5.7. Statistical Analysis

All results are expressed as the mean ± SD. The SPSS version 22.0 software (SPSS Inc. Chicago, IL, USA) was used for data analysis, the differences between the control and treatment groups were performed using a one-way analysis of variance (ANOVA) and an LSD test to reveal the differences between the two groups. *p*-values < 0.05 were considered significant.

## Figures and Tables

**Figure 1 toxins-14-00833-f001:**
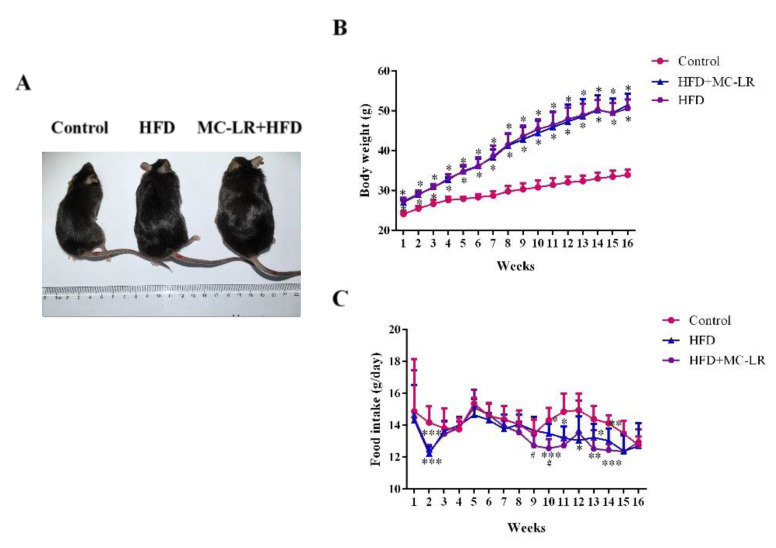
Effects of MC-LR on the body weight and food intake in HFD-induced obese mice. (**A**) The appearance of the animal; (**B**) Body weight gain was measured for 16 weeks; (**C**) Food intake in different groups. Data are presented as the mean ± SEM; *n* = 5. * *p* < 0.05, ** *p* < 0.01, *** *p* < 0.001 compared with the control mice; # *p* < 0.05 compared with the HFD mice.

**Figure 2 toxins-14-00833-f002:**
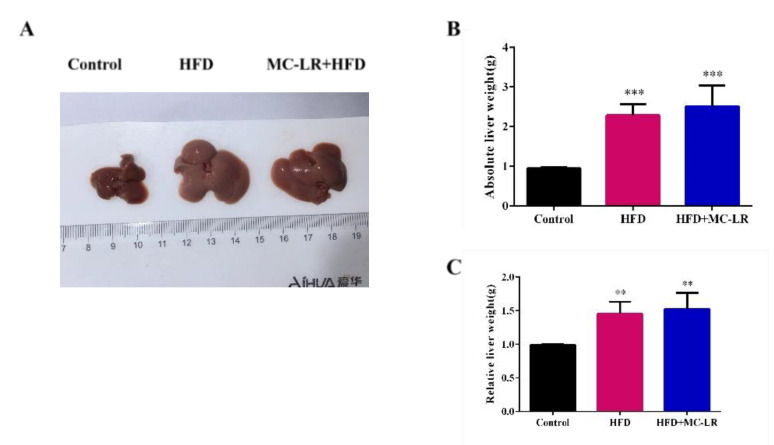
Effects of MC-LR on the liver index in HFD-induced obese mice. (**A**) The appearance of the liver tissue; (**B**) Absolute liver weight; and (**C**) Relative liver weight in different groups. Data are presented as the mean ± SEM; *n* = 5. ** *p* < 0.01, *** *p* < 0.001 compared with control mice.

**Figure 3 toxins-14-00833-f003:**
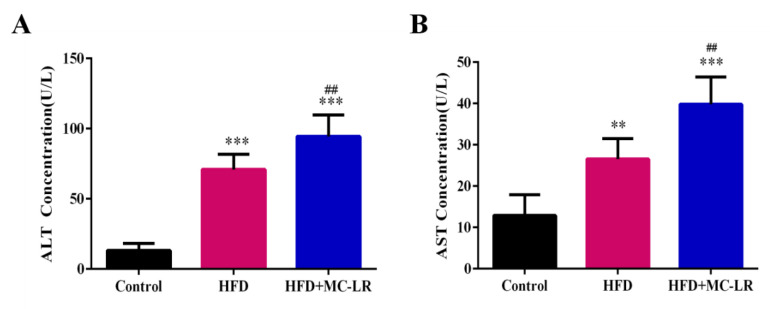
Effects of MC-LR on liver function marker enzymes in HFD-induced obese mice. (**A**) The concentration of serum ALT; (**B**) The concentration of serum AST. Data are presented as the mean ± SEM; *n* = 5. ** *p* < 0.01, *** *p* < 0.001 compared with control mice; ## *p* < 0.01 compared with HFD mice. Abbreviation: ALT: alanine aminotransferase, AST: aspartate aminotransferase.

**Figure 4 toxins-14-00833-f004:**
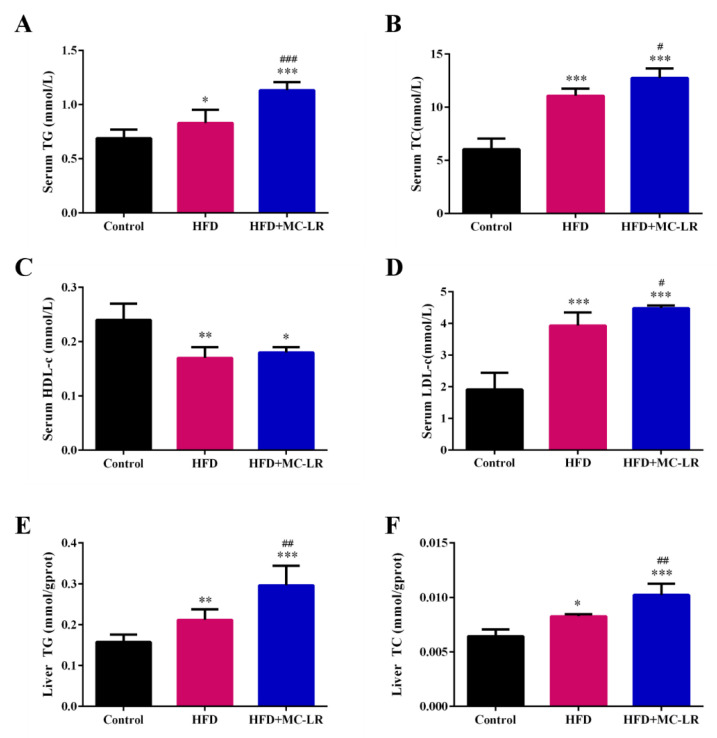
Effects of MC-LR on the serum lipid index (**A**–**D**) and liver lipid index (**E**,**F**) in HFD-induced obese mice. Data are presented as the mean ± SEM; *n* = 5. * *p* < 0.05, ** *p* < 0.01, *** *p* < 0.001 compared to control mice; # *p* < 0.05, ## *p* < 0.01, ### *p* < 0.001 compared with HFD mice. Abbreviation: TG: triglyceride; TC: total cholesterol; HDL-c: high-density lipoprotein cholesterol; LDL-c: low-density lipoprotein cholesterol.

**Figure 5 toxins-14-00833-f005:**
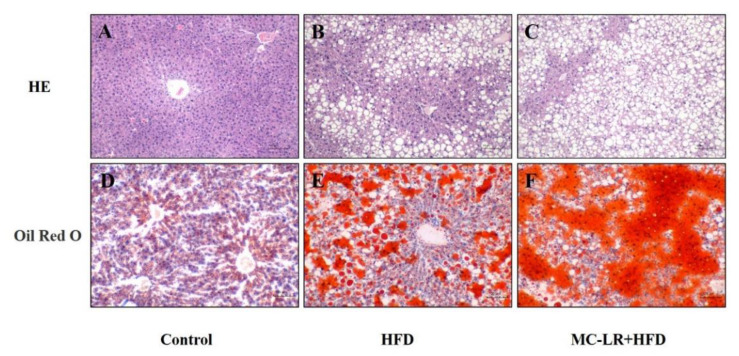
Histopathology analysis of liver tissues from the mice in different treated groups. (**A**–**C**) The structure of histological sections of liver was observed by H&E staining; (**D**–**F**) The structure of histological sections of liver was observed by oil red O staining. Bar = 100 μm means original magnification ×200. Abbreviation: H&E staining: hematoxylin and eosin staining.

**Figure 6 toxins-14-00833-f006:**
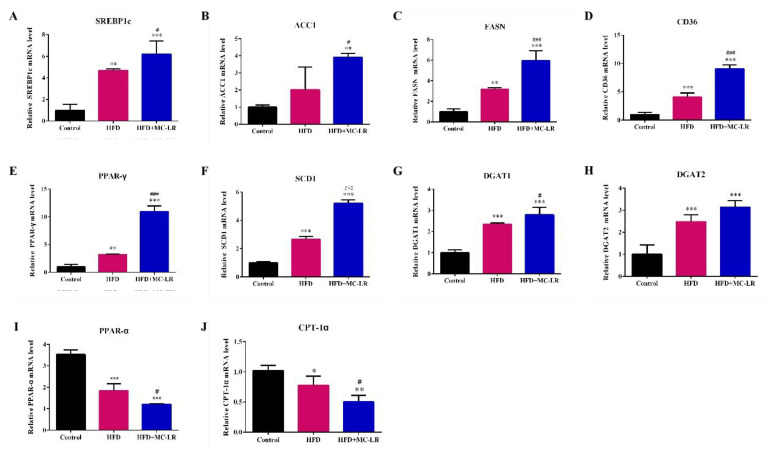
Effects of MC-LR on lipid synthesis and fatty acid β-oxidation-related gene expression levels in HFD-induced obese mice liver. (**A**–**H**) The mRNA expression levels of lipid synthesis-related genes; (**I**,**J**) The mRNA expression levels of fatty acid β-oxidation-related genes. Data are presented as the mean ± SEM; *n* = 5. * *p* < 0.05, ** *p* < 0.01, *** *p* < 0.001 compared with control mice; # *p* < 0.05, ### *p* < 0.001 compared with HFD mice.

**Figure 7 toxins-14-00833-f007:**
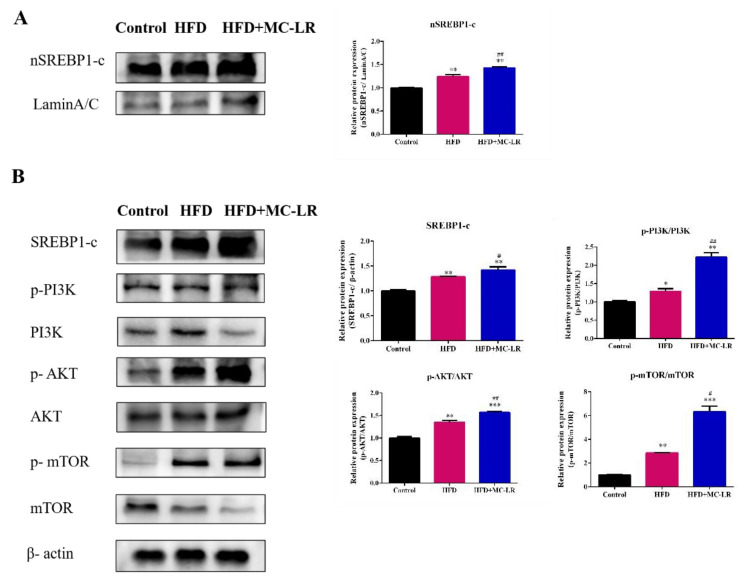
Effects of MC-LR on the PI3K/AKT/mTOR/SREBP1 pathway in HFD-induced obese mouse livers. (**A**) Western blotting analysis of protein nSREBP1-c, and relative quantitation of the protein level normalized to Lamin A/C; (**B**) Western blotting analysis of proteins (SREBP1-c, *p*-PI3K, PI3K, *p*-AKT, AKT, *p*-mTOR, mTOR), and relative quantitation of the protein level normalized to β-actin. Data are presented as the mean ± SEM; *n* = 5. * *p* < 0.05, ** *p* < 0.01, *** *p* < 0.001 compared with control mice; # *p* < 0.05, ## *p* < 0.01, compared with HFD mice.

**Figure 8 toxins-14-00833-f008:**
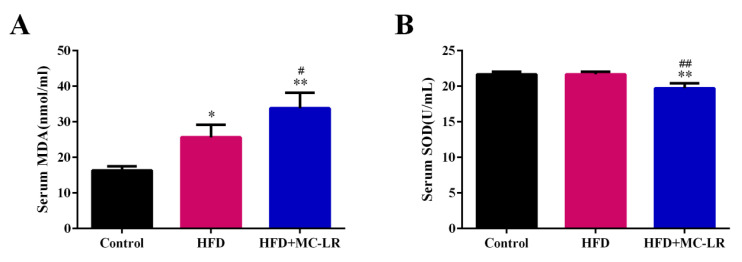
Effects of MC-LR on serum oxidative stress levels. Data are presented as the mean ± SEM; *n* = 5. * *p* < 0.05, ** *p* < 0.01 compared with control mice; # *p* < 0.05, ## *p* < 0.01 compared with HFD mice. Abbreviation: MDA: malondialdehyde; SOD: superoxide dismutase.

**Table 1 toxins-14-00833-t001:** Primer sequences for qRT-PCR.

Genes	Forward Primer (5′–3′)	Reverse Primer (5′–3′)
*SREBP-1c*	TGGAGACATCGCAAACAAG	GGTAGACAACAGCCGCATC
*DGAT2*	GGCTGGCATTTGACTGG	TGGTGGTCAGCAGGTTGT
*FAS*	GCCTCCGTGGACCTTATC	ACAGACACCTTCCCGTCA
*ACC1*	AAGGGACAGTAGAAATCAAA	CAGCCTCCAGTAGAAGAAG
*SCD1*	GGGAATAGTCAAGAGGCT	ACGAGGACGACAATACAA
*DGAT1*	GTGGGTTCCGTGTTTGC	CTCGGTAGGTCAGGTTGTCT
*PPARγ*	TTCGCTGATGCACTGCCTAT	TGATCGCACTTTGGTATTCTTGG
*CD36*	GGCAGGAGTGCTGGATTA	GAGGCGGGCATAGTATCA
*CPT1A*	ATGTTTCGACAGGTGGTT	TGCGTTTATGCCTATCTT
*PPARα*	AGGGCCTCCCTCCTACGCTTG	GGGTGGCAGGAAGGGAACAGA

## Data Availability

The data presented in this study are available in this article.

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
