# Peer review of "MC-LR Aggravates Liver Lipid Metabolism Disorders in Obese Mice Fed a High-Fat Diet via PI3K/AKT/mTOR/SREBP1 Signaling Pathway"

_toxins, 2022, doi:10.3390/toxins14120833_

Round 1
Reviewer 1 Report
Line 55: MC-LR is easily treated in drinking water with KMnO4 and chlorine in 10 minutes which is a standard drinking water treatment practice. https://doi.org/10.1016/j.watres.2007.10.039 It is also weird to cite for this statement, a review on immobilized bacterial communities for the general claim of “drinking water treatment practices for microcystins”, as if this is the only way to treat for them. It would be more helpful to site this and other papers regarding the heterogeneous bacterial communities that degrade MCs naturally, and then also the standard drinking water treatment practices.
Reviewer 2 Report
In this article, the authors aim to show how low-dose exposure to MC-LR affects liver lipid metabolism in high-fat diet-induced NAFLD conditions. Although the article is scientifically sound and the data presented supports the hypothesis, the data could have used an additional control group where mice were given a normal fat diet and MC-LR. I appreciate that the mice were given MC-LR in drinking water as opposed to I.P. injections, however, just for clarification please mention how often was the drinking water replenished with MC-LR.
The introduction and Discussion could be improved with references such as
1. Diez-Quijada, L., Benítez-González, M.D.M., Puerto, M., Jos, A. and Cameán, A.M., 2021. Immunotoxic effects induced by microcystins and cylindrospermopsin: A review. Toxins, 13(10), p.711;
2. Lad, A., Breidenbach, J.D., Su, R.C., Murray, J., Kuang, R., Mascarenhas, A., Najjar, J., Patel, S., Hegde, P., Youssef, M. and Breuler, J., 2022. As we drink and breathe: Adverse health effects of microcystins and other harmful algal bloom toxins in the liver, gut, lungs and beyond. Life, 12(3), p.418.
The article needs extensive English language and grammar checks along with punctuation and spacing checks. Also, a schematic would be more helpful to enhance the article.
Reviewer 3 Report
The manuscript deals with investigating the effects of MC-LR on the liver lipid metabolism of obese mice and clarifying the pathway by which these effects occur. It could constitute an interesting report, as there are no significant problems with its scientific soundness; however, its quality is severely compromised by the very poor usage of English language, making the manuscript almost incomprehensible in certain parts. To this reviewer’s opinion, the manuscript requires substantial improvements to become comprehensive and reader-friendly. Detailed points for revision follow:
Points for Revision:
General points:
- The use of English needs very substantial improvement, so the authors are strongly advised to have the manuscript corrected by a native English speaker (or use a relevant professional service) before resubmitting a revised version. The problem is very evident in all sections of the manuscript, but is much more pronounced in the Abstract, Introduction and Conclusions sections, which are almost unreadable, with a completely chaotic syntax. In fact, the text in many parts seems like it has been produced by an automatic translation software, as many words are misplaced in the sentences or used in a wrong tense.
- There is a general issue with the use of unexplained abbreviations. The authors should carefully re-read the manuscript and provide the explanation for all used abbreviations at their first instance in the text (some are explained much later, probably due to the fact that the Materials and Methods section in Toxins is placed towards the end of the manuscript). Also, the use of abbreviations should be explained in all relevant figures, as the figures should be able to stand alone, without the reader needing to look in the main text in order to understand what an abbreviation means.
Other specific comments
Title:
- Correct “signaling” to “signalling”.
Abstract:
- Please rewrite the whole abstract; there is not a single phrase without a confusing syntax.
1. Introduction
- Same as in the abstract, almost the whole introduction section requires re-writing in terms of language (there is no significant problem with the actual contents of the text, but the reader cannot follow it because of the syntax).
- Page 2, lines 56-57: The guideline values have been recently revised by WHO (in 2020); it would be better to have the recent reference and indicate the newly established levels (different for short and long-term exposure).
- Page 2, line 65: “exposure” appears twice, delete one instance (also after revising the text).
2. Results
- Page 4, lines 105-119: All instances of “figure” should be capitalized.
- Figures 3, 4, 5: explain the abbreviations.
3. Discussion
- Page 10, lines 193-204: This whole paragraph should be deleted, it’s a repetition of information provided in the Introduction section, which is unnecessary here.
- Many parts with confusing syntax, please check.
4. Conclusions
- Same as in the abstract and Introduction, almost the whole conclusions section requires re-writing in terms of language (there is no significant problem with the actual contents of the text, but the reader cannot follow it because of the syntax).
5. Materials and Methods
- Page 13, line 334: Please comment on how adequate MC-LR dosing was secured for each mouse? Were the mice housed individually or in groups? How could the intake of the toxin be estimated?
Round 2
Reviewer 3 Report
All comments relevant to scientific issues have been well addressed. The use of English has slightly improved, but still many phrases are syntactically incorrect (e.g. Abstract: lines 5-9, 13-16, Introduction: lines 63-65, etc.). Some further English correction is required before the manuscript becomes publisable.